

# Characteristics of Global Oceanic Rossby Wave and Mesoscale Eddies Propagation from Multiple Datasets Analysis

Yunfan Zhang[1], Fenglin Tian[1,2], Ge Chen[1,2]

[1]Qingdao Collaborative Innovation Center of Marine Science and Technology, College of Information Science and Engineering, Ocean University of China, Qingdao, 266100, China
[2]Laboratory for Regional Oceanography and Numerical Modelling, Qingdao National Laboratory for Marine Science and Technology, Qingdao, 266100, China
*Correspondence to*: Fenglin Tian(tianfenglin@ouc.edu.cn)

**Abstract.**

In this paper we present a research of propagation characteristics of global Rossby wave and mesoscale eddies, and we preliminarily discussing the relationship between them from multiple datasets analysis. By filtering the MSLA-H data and by means of optimized SSH method we have extracted signals of the Rossby wave, and estimated the propagation speed (zonal phase speed) of the Rossby wave and eddies. Validation for the identification of the Rossby wave also has been completed with the Argo temperature and salinity data. The prime focus covers: propagation speed comparison between the Rossby wave and the eddies, propagation characteristics in different regions. Overlaying the signals of the Rossby wave with the signatures of the eddies indicates that the Rossby wave and the eddies propagates together (westward only) in the mid-latitude, but differences appear with increasing of latitude, especially in some areas affected by ocean current, for instance, the West Wind Drift(WWD) and the North Atlantic Drift(NAD). Actually we have found that the currents led the eddies, and the Rossby wave might play an accelerative or moderative role in the eddies propagation, as a result of the velocities of the eddies and the currents were matched well, but comparison between the Rossby wave and the eddies revealed disparity. The findings are useful for understanding the relationship between the Rossby wave and mesoscale eddies.

## 1 Introduction

Oceanic Rossby wave and mesoscale eddies play an important part in the dynamic ocean, and an increasing number of oceanographers have concentrated on both of them, which are ubiquitous but invisible phenomenon in global ocean, and many researches have been proceeded for finding the evidence of existence of them. Using MSLA-H data, the method adopted in this paper mainly differentiates the eddies and the waves at different latitudes and regions, and makes verification for the results by SST and Argo data.

Rossby waves are planetary waves that are supported by the Earth's rotation through the latitudinal dependence of the Coriolis parameter (Gill, 1982), and a number of different "modes" can exist with different depth structures (Quartly et al., 2000). The barotropic mode travels too fast to be observed by the satellites, so almost all studies focus on the first baroclinic mode, and this paper as well. From sea level signals with approximately <10-centimeter amplitude and >500-kilometer wavelength, (Chelton *and Schlax*,1996) have presented Rossby waves throughout much of the world ocean by using TOPEX/POSEIDON data.



(Cipollini et al., 1997) have estimated the wavelength, period and propagation speed of Rossby waves near 34°N in the Northeast Atlantic by using T/P SSH data and ERS-1 SST data. (Challenor and Cipollini et al., 2004) have calculated the T/P derived Rossby

wave propagation speeds and compared the speeds with the speeds predicted by the theory of (Killworth et al., 1997), as expanded in (Killworth and Blundell, 2003). We also have estimated the phase speeds of Rossby wave at different latitude by plotting the longitude/time Hovmöller diagrams, although the speeds are the dominant phase speed of westward SSH propagation and does not distinguish between contributions from different processes, including the waves and eddies etc. (Barron et al., 2009). To validate the westward propagation of barrier layer formation in tropical Southwest Indian Ocean, (Chowdary et al., 2009) analyzed Argo

temperature and salinity anomaly profiles in 2006-07 Rossby wave event. It proves that the waves can interact with the surrounding ocean phenomenon, especially related to temperature and salinity.

With the development of the marine satellite remote sensing technology, marine remote sensing data is increasingly applied to the study of the phenomenon of oceanic mesoscale eddies since the 1990s, and many algorithms have been devoted to eddy identification. (Xiu P et al., 2010) used Okubo-Weiss(OW) technique to investigate the features of vortex based on the MSLA-H

and ROMS datasets, and they tracked the eddies by using pixel connectivity algorithm. Wingding-Angle(WA) method and Vector Geometry(VG) method, are also put in use for the eddies, which have been reported in the research of (Chen G et al., 2011; Liu Y et al., 2012).

Furthermore, (Chelton et al., 2007; 2011) analyzed the distribution of global linear and nonlinear long life cycle mesoscale eddies by using OW and satellite altimeter method, and the movement of the eddies by using nearest neighbor method. To further

understand the characteristics of the eddies, statistical analysis has been made by (Chaigneau et al., 2008), and they have traced the eddy tracks by using similarity based algorithm (Penven et al., 2005). In order to reduce the error recognition, (Mason E et al., 2014) added vortex shape test and limitation of local extrema to (Chelton et al., 2007). For summarizing and discovering the pattern of eddy propagation, (Faghmous et al., 2015) established the 1993-2014 identification and tracking datasets. Based on the improved SSH method, we also established 1993-2015 eddy identification set.

The characteristics of satellite altimetry are as follows: 1) it can overcome the limitation of time and space, and offer an integrated overview of the propagation trends over the Rossby wave, mesoscale eddies, and other processes; 2) its dataset is easily obtained after several decades' accumulation, and that is very necessary for tracking large-scale phenomena, such as the Rossby wave and mesoscale eddies, especially when we try to make them distinctions; 3) database of daily, globally gridded MSLA-H (or SST) field can be reconstructed and analyzed advantageously. (Krieger and Polito, 2013) analyzed the spectral variability of the

filtered sea surface height (SSH) fields associated to first mode baroclinic Rossby waves using wavelet analysis, and altimetry data analysis makes measurement of the Rossby waves' characteristics (speed, wavelength, period) possible (Challenor et al.,2004). Works on identification also have been presented by using ocean color data (Cipollini et al.,2001) and ATSR SST data (Hill et al., 2000).





The major goals of this paper are as follows: 1) to introduce a database of daily, globally gridded MSLA-H and SST analyses,

2) to provide the method of data processing to make a comparison between the waves and the eddies, and 3) to present the

characteristics of Rossby wave and mesoscale eddies propagation at different latitudes and regions, and preliminarily discussing

the relationship between both of them. Such an immediately method is limited to zonal propagation analysis.

The organization of this paper is: The section 2 mainly introduces the data we selected and the application we decided. The

coverage and processing applied to prepare MSLA-H, SST and Argo S/T data is reported in section 3, including the feature

detection, series of Hovmöller diagrams plots of MSLA-H and SST data, and series of anomaly profiles of Argo S/T data.

Comparisons and issues associated with the characteristics of the waves and the eddies are introduced in section 4. More

discussion is contained in section 5, and the conclusions summarize results in the context of prior studies.

**2 Data**

Since Aviso has been distributing satellite Topex/Poseidon in 1992, launches of new altimetric mission followed have been

being built up series of products. Combining available data of two satellites, features and time availability of all available products

are obtained now. We have access to a delayed time gridded multimission product Maps of Sea Level Anomaly (MSLA-H and

MSLA-UV). Benefiting from the Physical Sciences Division established in 2005 of the ESRL, we have access to the NOAA

High-resolution Blended Analysis of Daily SST and Ice. China started Argo plan since 2002. With the data sharing from global

Argo floats, China Argo Real-time Data Center assimilated and regridded the data on multiple degree and multiple scale grid. In

this case, Quality controlled BOA Argo temperature and salinity data from CARDC will be applied.

In this study we used data from Aviso, NOAA and CARDC, and they all cover more than 10 years. For Aviso we used all

satellites merged delayed time product MSLA-H and MSLA-UV. The daily mean data have computed with respect to a twenty-

year mean, and have been meshed onto a 0.25°×0.25° global grid. For NOAA we used high resolution OI of SST using the

AVHRR data, which has reported in (*Reynolds et al.*, 2007). This daily mean data has been regridded onto a 0.25°×0.25° global

grid. For CARDC we downloaded the BOA Argo S/T data, and the monthly mean data is on a 1°×1° grid. Because of the various

data with different commencement date, all data we used above cover 5 years from 2004 to 2009.

Besides, the objectively analyzed climatology World Ocean Atlas 2009 (WOA2009, monthly, 1°×1° grid) data, and zonal

velocity of OGCM for the Earth Simulator (OFES, monthly, 0.1°×0.1° grid) data is also used in this study.

**3 Methods**

**3.1 Extraction and Verification of signals of Rossby wave**

In consideration of the Rossby wave mainly propagates westward (Chelton and Schlax,1996), the series of daily MSLA-H

fields are adopted for detecting the westward propagation signals. This signals may consist of various oceanic phenomenon, such

as eddy migration, Rossby wave diffusion, and other mesoscale dynamics. In this study we already had the results of eddy

identification, and it's easily to distinguish the real features of Rossby wave by the comparison between the results and the signals.



95 The signals of Rossby wave are preliminarily acquired with the MSLA-H datasets, and we have established a collection of longitude-time plots at every given latitude (5 degrees' interval, range from 60°S to 60°N at integer latitudes). Propagating waves emerge as diagonal features in the plots, which are considered as the first guess features. Figure 1 shows the longitude-time plot at 25°N in the &North Pacific Ocean. The most obvious peculiarity is the variations of annual stripe structure of height anomalies. With detailed observation on the anomalies, weak or strong signals are propagating to the west, but it is clear that some filtering is

100 needed in order to reduce high frequency noise and make these distinct. Figure 2 gives the same plot after low-pass filtering (running average by using a 30-day boxcar filter) along the time direction only, and then removing the zonal average. It's obviously that the high frequency noise has been removed and the signals of Rossby wave are more explicit and clear than the figure 1. To avoid edge effects, the filter was applied to a plot 30 points (7.5°) narrower (15 points on each side) than shown in figure 1, and 30 pixels were removed from each side after filtering.

105 For a latitude section plotted as figure 1 with the rightward increase of longitude and upward increase of time, the westward propagation signals produce patterns sloping upward from left to right (e.g., Dunkerton and Crum, 1995). Westward propagation is evident in the patterns within figure 2, where the 6 solid diagonal lines are drawn according to the subjective determination of an image analysis, as in Barron et al. (2009). To validate and supplement the results obtained from the subjective analysis within figure 2, figure 3 gives the SST longitude-time plot in the same time series and latitude as mentioned before, and figure 4 shows

110 the filtered SST using the same filtering approach with the SLA analysis. The interannual variation which peak value emerges in every second half year and valley value appears in every first half year occupies the predominant feature, and weak stripe structure of temperature also appeared in the SST field. After being filtered the same as SLA (the SST field has been normalized first), the stronger or weaker westward features of the image than the figure 2 can be confirmed. There are also 6 dotted sloped lines that have been appeared at the same position of the solid lines of the figure 2 drawn by subjective method, and in addition, a

115 solid-dotted line is seen as a complementary signal for the figure 2.

 Furthermore, due to the propagation of the Rossby wave causes the variation of the thermocline (Chelton and Schlax,1996), we also take the Argo S/T in to consideration for tracking the signals belong to the Rossby wave. Following (Killworth et al., 1997), and as also explained by Barron et al. (2009), speed of the Rossby wave in the 25°N is about 5 cm/s, hence the anomalies on the thermocline migrate 1° per 28.01 days in zonal direction. Here we adopt interval of 3 months to show the changes of

120 thermocline. Figure 5 f) shows the longitude-depth of monthly Argo-S and Argo -T section series in the same area noted by the lines mentioned before. The shaded part represents the anomaly by real Argo S/T minus the WOA 2009 S/T, and the contours are the real Argo S/T. As shown in figure 5 all the lines depicted in figure 2 and figure 4 (refer in particular to temperature complementary line, TC) are appeared by reflecting in thermocline anomalies propagation, as the arrows pointing. Although the anomalies not always arise in the pictures they should emerge or show blow the thicken 20°C contour, the fast or slow spread of

125 westward trend is clear and definite. The features of salinity anomalies in figure 6 are similar to the characteristics of temperature





anomalies in figure 1e), and lines shaded with the color marked the signals in figure 4 also show the trend of the west.

With the results above we estimated the phase speed of propagation of the Rossby wave. The formula for calculating the speed is given. In this case we assume the meridional wave number $k_y = 0$, that is,

$$c_x^p = \frac{\sigma}{k_x^2} = \frac{\cos\theta}{R} \approx \frac{1}{R}$$

where R is the reciprocal of the slope of the 7 lines noted in the text, and $\theta$ is the included angle between direction of wave propagation and latitudinal direction. From the 7 samples we estimated the phase speed of the Rossby wave at 25°N in the north Pacific Ocean is 4.68cm/s.

Figure 1

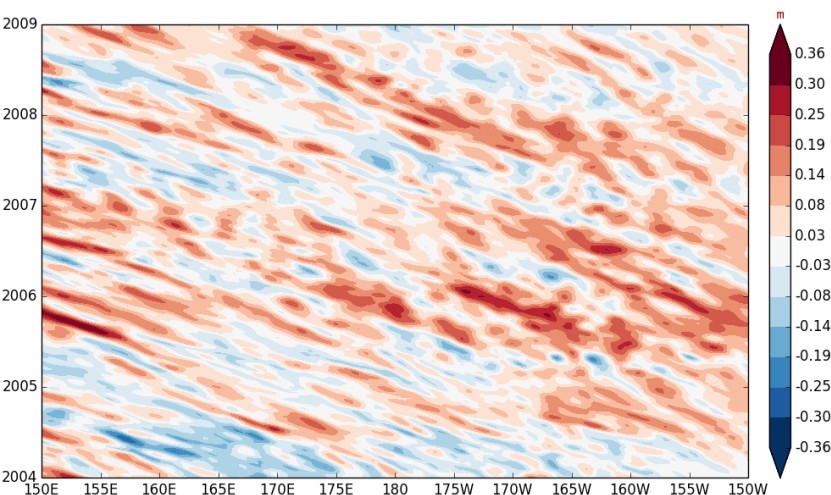

The longitude-time plot of real SLA at 25°N from 2004 to 2009. Interannual variations of stripe structure of SLA show the trend from west to east, with the passage of time. This real SLA field is extracted from the daily MSLA-H time series, as mentioned in the text.

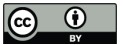



Figure 2

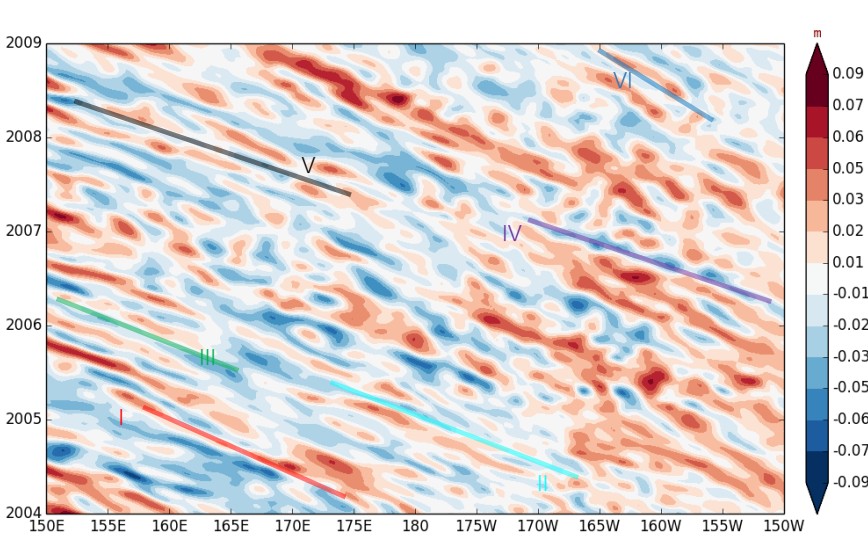

The same plot as figure 1, but a 30-day boxcar filter has been applied to the SLA field, and zonal average has been removed. Note

that most of the high-frequency noise has been removed, and the areas full of 'spotted structure' in figure 1 become clear and the

plot is more suitable for analyzing. The 6 solid lines which probably represent the probable propagation of the Rossby wave have

been extracted based on the subjective determination of image analysis.

Figure 3

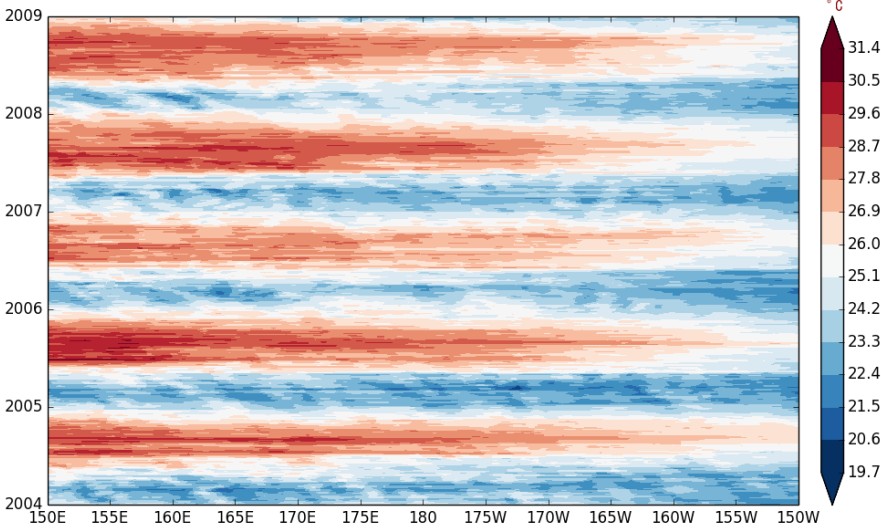

The real SST longitude-time plot same as SLA field at 25°N from 1994 to 2003. Trends of warm and cold vary with the seasonal

variation, but many westward anomalies can be clearly observed when we zoom in for details (not show here). This real SST field

is extracted from the daily OI of SST series, as mentioned in the text.


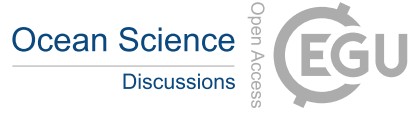

Figure 4

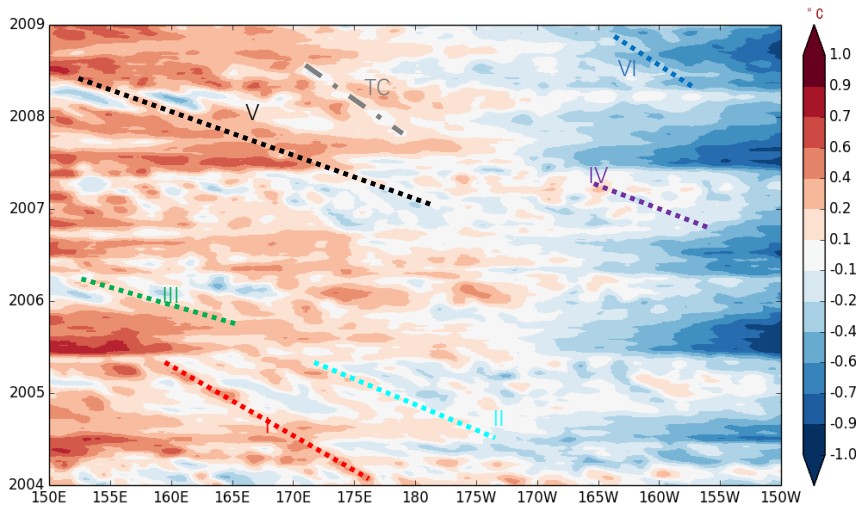

The same plot as figure 3, but the SST field has been normalized in range -1 to 1, and 30-day boxcar filter has been applied to the SST field, and zonal average has been removed. 6 dotted lines like figure 2 can be the evidence for validating the signals which

belong to the Rossby wave. Otherwise, a line which not being identified in the figure 1b) appears, and it is also treated as the probable signal of propagation of the Rossby wave.

Figure 5

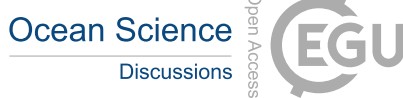











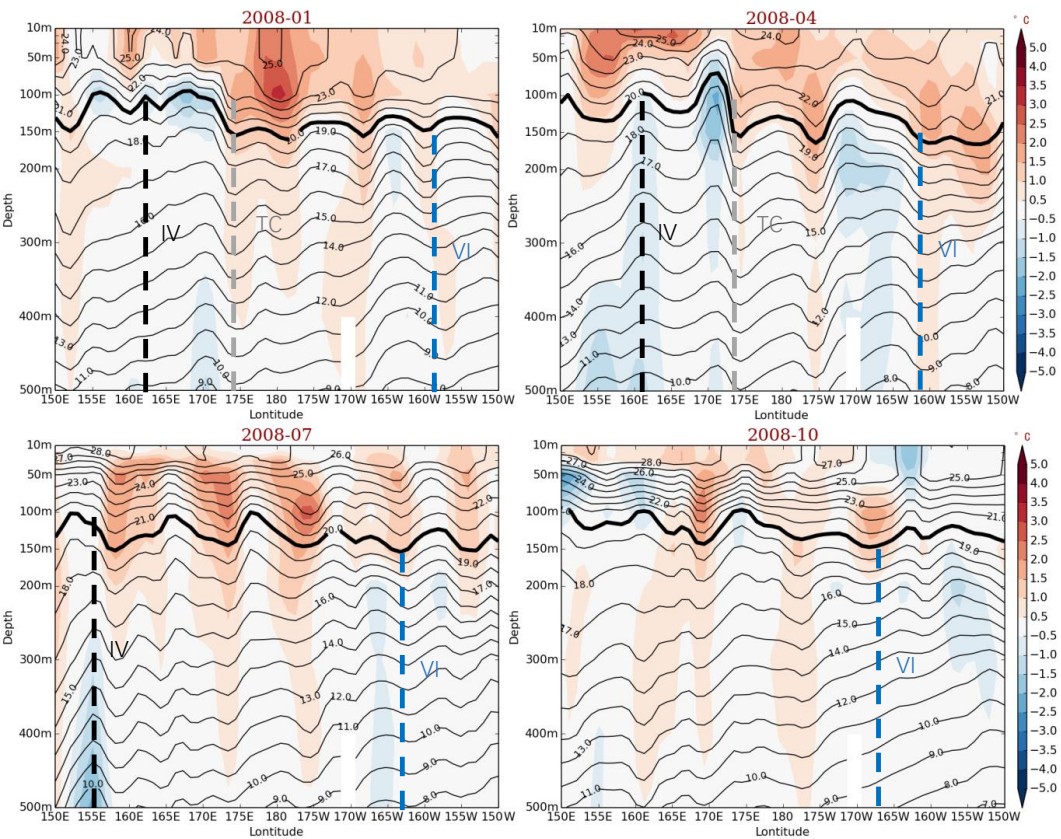


Longitude-depth sections of interval of 3 months Argo subsurface temperature anomalies (shaded in °C), superimposed on real

temperature (the 20 °C isotherm thickened) at 25°N in the north Pacific Ocean from January 2004 to October 2008.



Figure 6











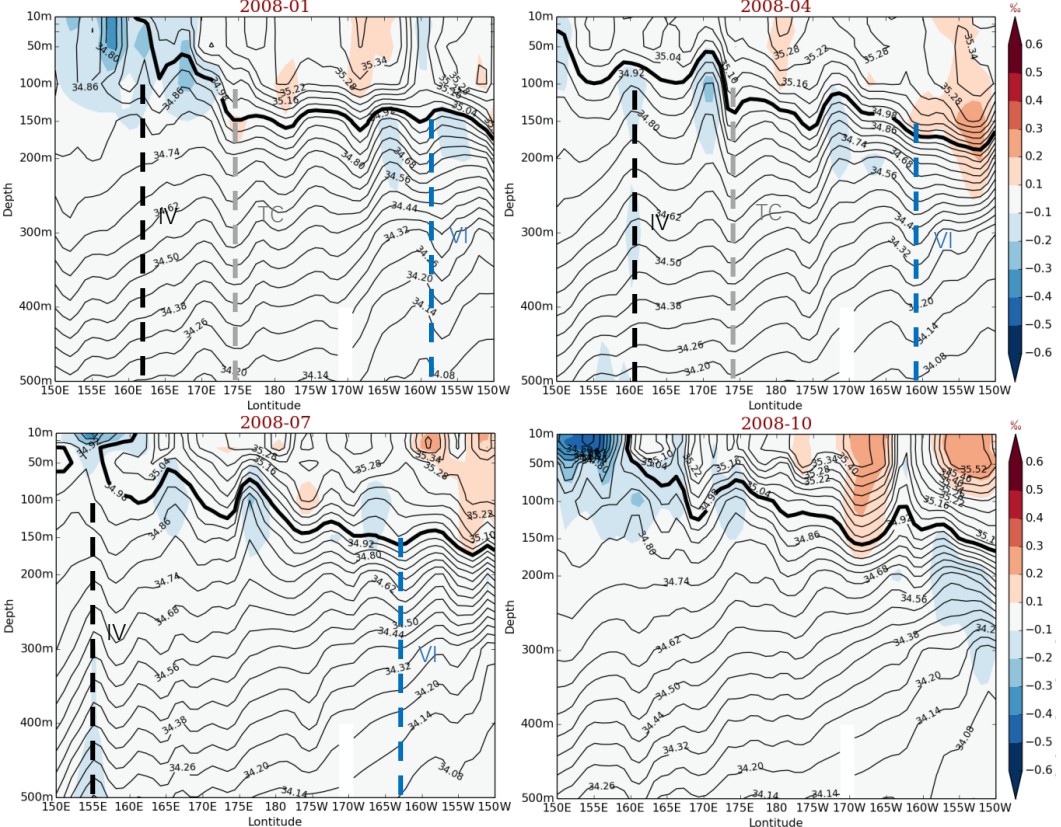

Longitude-depth sections of interval of 3 months Argo subsurface salinity anomalies (shaded in ‰), superimposed on real salinity

(the 34.98‰ contour thickened) at 25°N in the north Pacific Ocean from January 2004 to October 2008

### 3.2 Phase speed estimation of the Rossby wave and eddies

Time series of daily MSLA-H fields are used to examine the Rossby wave propagation of the anomalies range from 60°S to

60°N at 5 latitudes intervals, and we also estimated the phase speed of the Rossby wave. Following (Killworth et al., 1997), the

predicted phase speed of nondispersive Rossby waves neglecting background currents is $-\beta C_1^2/f^2$, where $f$ is the Coriolis

parameter, $\beta$ is the rate of change of $f$ with latitude, and $C_1$ is the mode 1 internal wave speed, which has interpolated from

estimates of first baroclinic Rossby wave phase speed (Chelton et al., 1998). Figure 7 gives the global distribution of phase speed

of the Rossby wave. Samples concentrated between 10 degrees and 40 degrees of the equator of the Rossby wave in figure 7

reveal that prominent features tend to be more prevalent in the area without strong advection than in the area with strong

advection (Cipollini et al.,2001; Barron et al., 2009), and the abnormal values emerged at 15°S may be caused by the perturbation

of the Benguela Current in the Atlantic.

By analyzing the results of the mesoscale eddy identification we also estimated the global zonal phase speeds of mesoscale

eddies at the same locations where we have extracted signals of the Rossby wave. The mean zonal phase speeds are calculated by




averaging the speeds of eddies which emerged more than 60 days. Figure 8 shows the trend of zonal eddy propagation and figure

9 is the distribution diagram of the mean zonal phase speeds of eddies in different oceans. In fact, we also found that there were

many of eastward signals of eddies at high latitudes, and their phase speeds are much higher than the westward signals, especially

in the area affected by strong ocean currents.

Figure 7

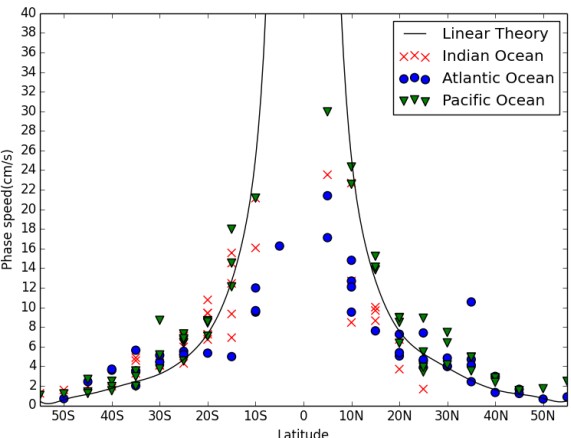

Comparison between the MSLA-H derived speed in different oceans and the global predicted speeds of the linear theory

(Killworth et al., 1997).

Figure 8

a)

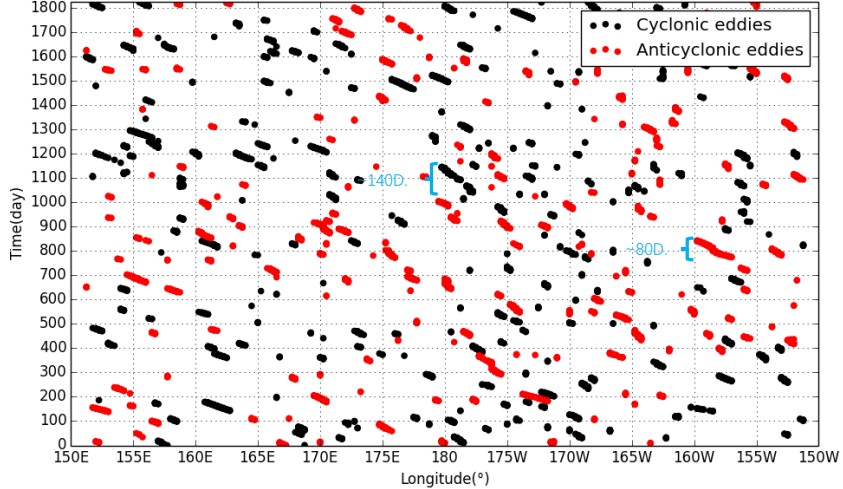

b)




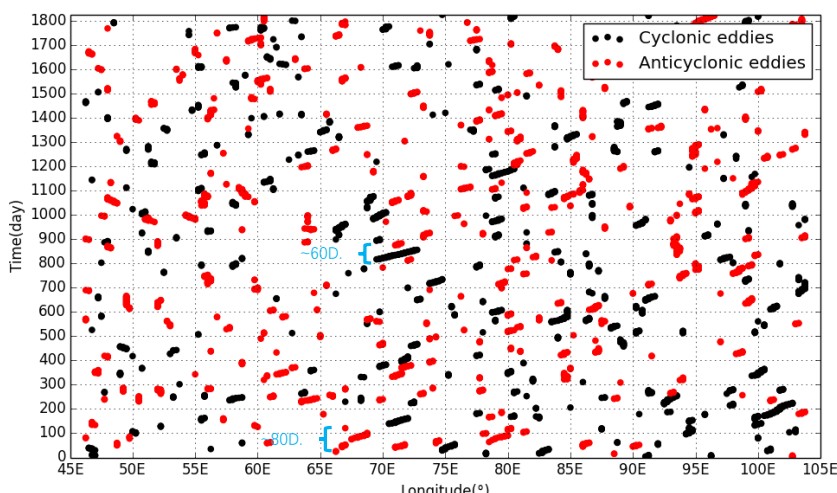


Longitude-time plot for visualizing the trend of zonal eddy propagation at 25°N in the Pacific Ocean (figure 8a) and at 45°S in the Indian Ocean (figure 8b) from 2004 to 2009. The dark and red spots represent the cores of the eddies emerged at the same given latitude but a different day of cyclonic and anticyclonic eddies, respectively. It's clear that both the cyclonic and anticyclonic eddies migrate from the east to the west in figure 8a, but it's just the opposite in figure 8b.

Figure 9

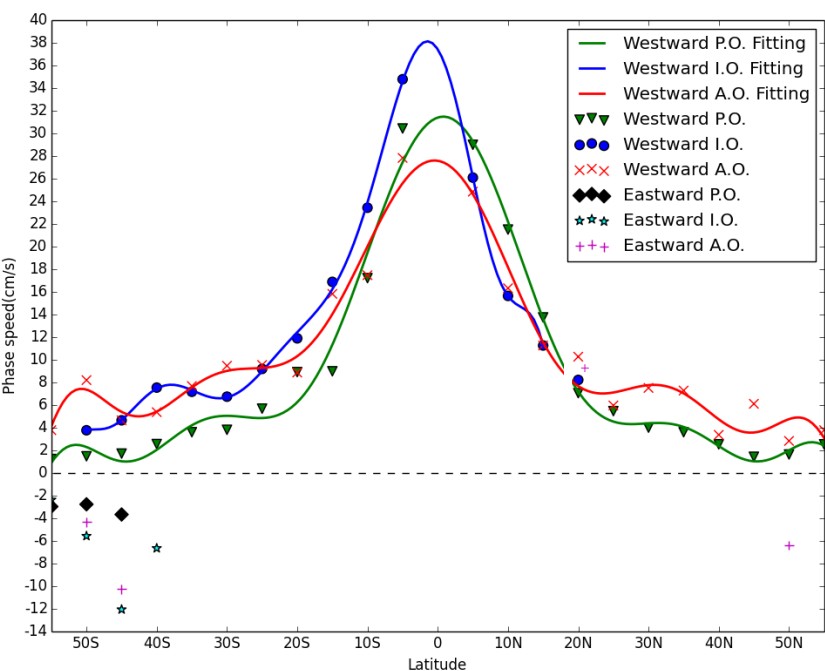





Diagram of scatters of the mean zonal phase speeds of eddies and polynomial fitting curves of points represent westward propagation eddies in the Pacific Ocean, Indian Ocean and Atlantic Ocean, respectively. Due to the boundary effects encountered in calculating the curves we did not chart the connection between the 15°N and 20°N on the Indian Ocean curve out. The eastward speed at 45°S is obviously faster than the westward speed in any ocean.

**4 Disscusion**

Many of studies have focus attention on the Rossby wave or mesoscale eddies, whereas few of research have developed the relationship and interaction between the waves and eddies. In this study we preliminary investigate the interrelation mentioned above. We have extracted the signals of the Rossby wave and estimated their zonal phase speeds at different latitudes, as discussed above. The result that features obtained from the MSLA-H fields correspond to the linear theory makes us believed that the features are real signals of the Rossby wave. Figure 10 shows the mean zonal phase speeds of eddies superimpose the theoretical speeds of the waves, and points represented the eastward propagation features of eddies in different oceans. Only a few coincident points between the curves appear, especially in the low latitudes Pacific Ocean. Note that near the south latitude 45 degrees in the Indian Ocean the disagreement reaches a peak, and the numerical value of the peak is close to 18°S, which can be proved in figure 11 as emphasized with black circle.

Figure 12 shows the change of the zonal geostrophic velocity at 45°S in the Indian Ocean from 2008 to 2009. It's clear that eastward trend is much stronger than the westward trend. We also calculated the average of the eastward zonal velocity and the value is -0.136 m/s. The value is close to the -0.121 m/s of the peak described in figure 10 (the value of phase speed of the Rossby wave at 45°S in the Indian Ocean is 0.011 m/s), and it proves that the propagation of eddies influenced by powerful horizontal flow follows the characteristics of the local advection, in another words, the ocean current is the primary driving force of the eddies, and the Rossby wave may have effect of dispersion. We have found this peculiarity not only in the WWD but also in the NAD. The velocity of the eddies at 50°N in the Atlantic Ocean is -0.064 m/s, which is fitted with the velocity -0.087 m/s derived from MSLA-UV.

Actually almost all of the oceans are influenced by the strong currents except the low latitudes Pacific Ocean, especially between 20 degrees and 30 degrees of the equator. Black triangle in figure 11 presents the area which the zonal velocity is weak at 25°N in the central Pacific Ocean. We take the distribution of meridional geostrophic velocity at 25 degrees of the equator as examples depicted in figure 13. In these areas the curves of the meridional geostrophic velocity migrate from the east to the west, which reveals that the eddies mainly move to the west. The averaged westward speeds of MSLA-UV are 0.093 m/s at 25°N and 0.117 m/s at 25°S, according with 0.072 m/s and 0.075 m/s from figure 10(the phase speed of the Rossby wave at the same location is 0.047 m/s and 0.046 m/s), and all the evidence above make us believed that the currents lead the eddies to move (in the zonal direction). It indicates that the Rossby wave may produce obstructive effects in most cases.

One more noteworthy thing is that the speeds are not constant at different locations and different time. The waves may speed



up or slow down (*Cipollini et al.*, 2001), as the figure 5 and 6 shown. The eddies also emerge that appearance as figure 8 shown. One explanation for this phenomenon can be the variation of the sea water state, and the influence factors may include the ocean

currents, and the local wind field.

Based on figure 2 and figure 8a, we give figure 14 to show the signals of the Rossby waves superpose the cores and effective radius of the eddies, and we assume that the six rectangles in the figure are the range of time and longitude influenced by the Rossby wave. We extracted the data included in the 6 rectangles from the whole field of results of eddy identification, and figure 15 gives the detailed longitude-time plot of the 6 areas for the eddies at 25°N in the central Pacific Ocean.    Table 1 gives the

average of zonal westward propagation speed for every area. Compared with the averaged zonal speed 5.43 cm/s in figure 9 (the average westward zonal speed value from MSLA-UV is 9.25 cm/s), most of the average speed of eddies in the rectangles are lower, but the values are close to the propagation speed of the Rossby wave (4.68 cm/s) at 25°N in the Pacific Ocean. Table 2 shows the statistics of variation of eddy effective radius and amplitude (the areas are same as table 1), and the ratio between the total number of effective radius extending eddies and effective radius decreasing eddies is 30:41, the ratio between the total

number of amplitude increasing eddies and amplitude decreasing eddies is 39:33. To investigate more information, we make a comparison between the cyclonic eddies and anticyclonic eddies in figure 16. We found that Cyclonic eddies change little, but the anticyclonic eddies change a lot. It seems that the Rossby wave may play an accelerative role in the process of the eddy propagation, and have some influence on the form of the anticyclonic eddies.

Figure 10

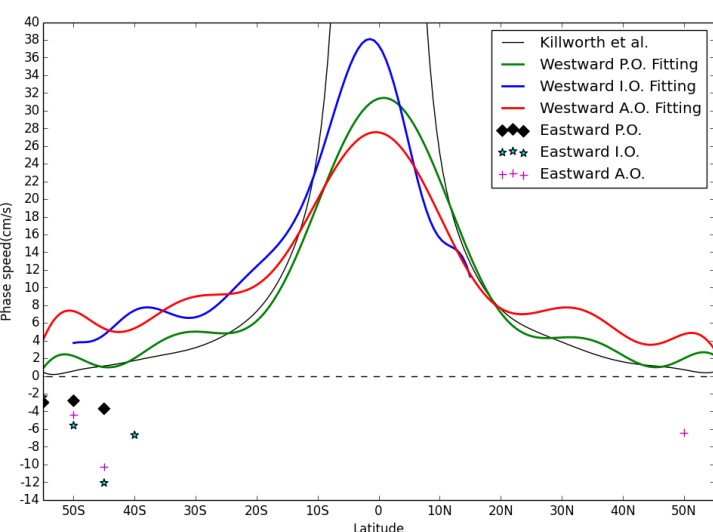


The same plot as figure 9 but removed the original data of curve fitting and superimposed the curve of predicted phase speed of the Rossby wave by Killworth et al.





Figure 11

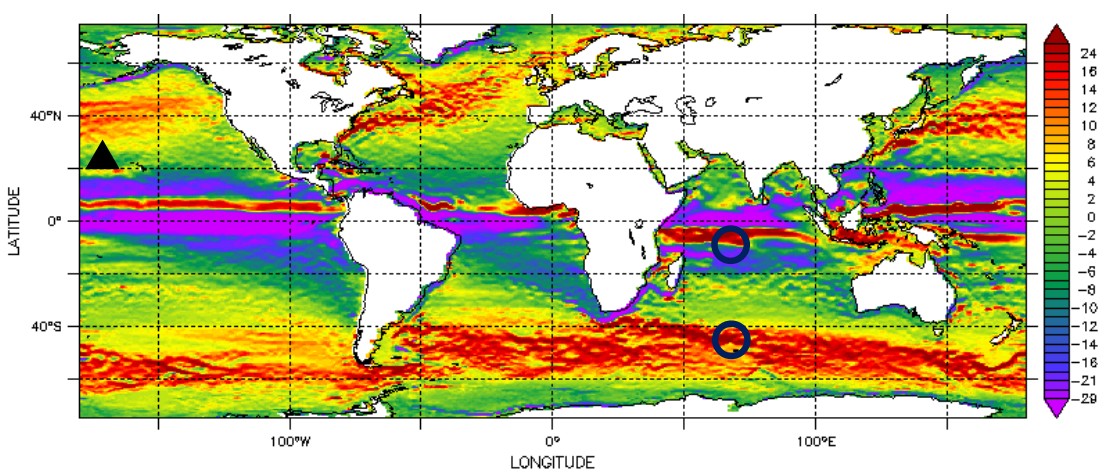

Global zonal velocity (no units) distribution of sea surface from OFES 10-years climatology dataset. The velocities in the area of

strong ocean currents are almost no change, hence we cite the January data in this case.

Figure 12

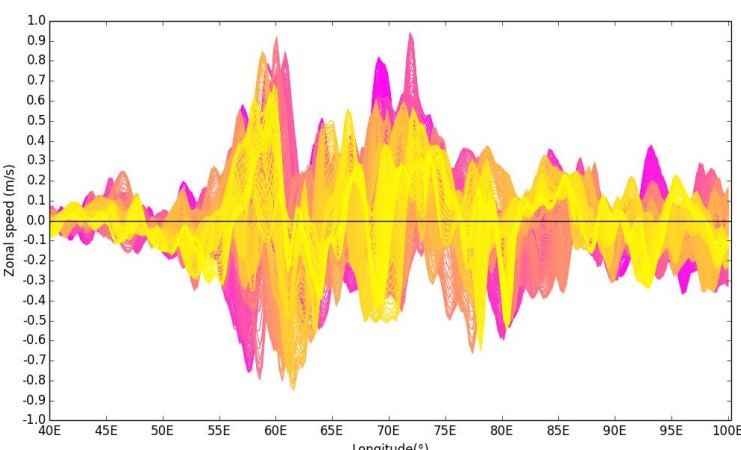

Variation of zonal geostrophic velocity derived from a 2008-2009 MSLA-UV field at 45°S (in the Indian Ocean). The color

changes from purple to yellow means the time varies from 2008 to 2009. The positive value and negative value represent the

westward and eastward velocity, respectively, and the average of eastward speed is -0.136 m/s.

Figure 13

a)



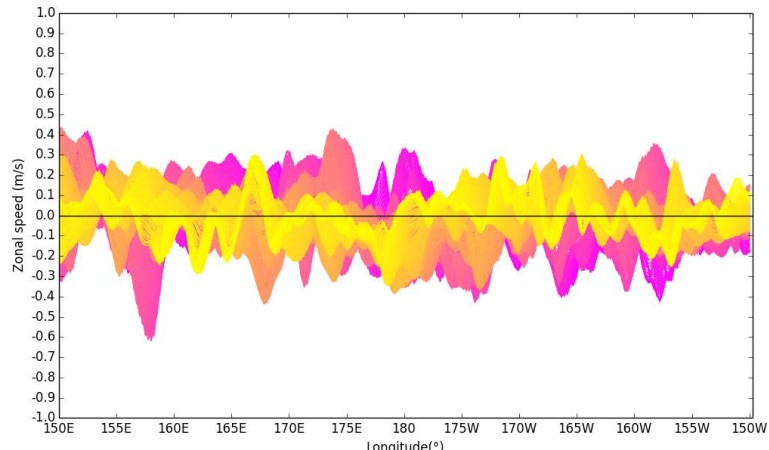

b)

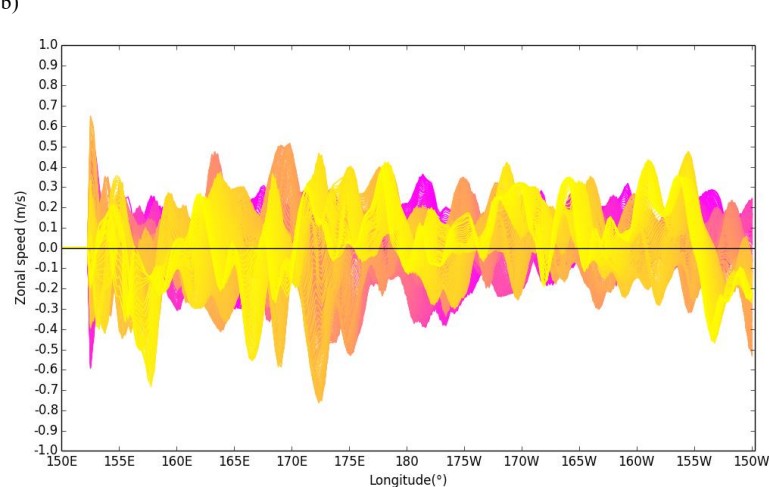

Variation of meridional geostrophic velocity derived from a 2008-2009 MSLA-UV field at 25°N(a) and 25°S (both in the Pacific

Ocean). The color changes from purple to yellow still means the time varies from 2008 to 2009. Peaks of the speeds shows the

westward migration trend with the passage of the time.

Figure 14



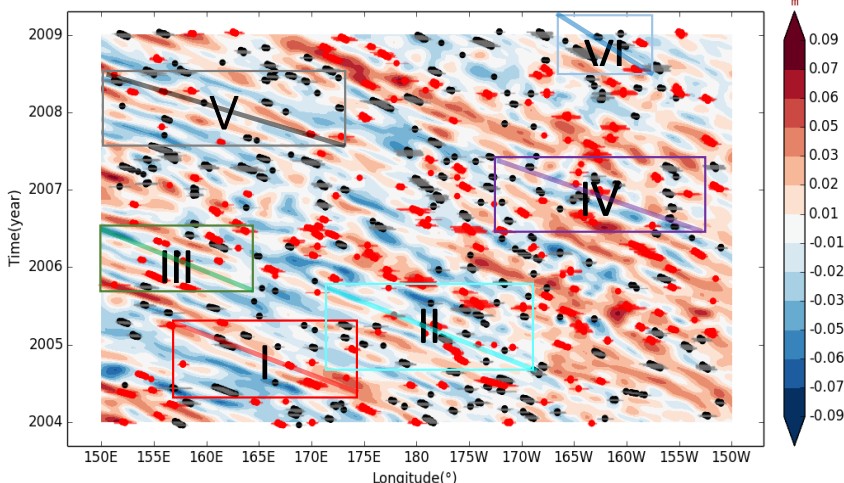

Diagram of superposition of the figure 2 and the figure 8a. The red or gray lines across the red or black points which represented

eddies are the effective radius of the different eddies.

Figure 15

a)

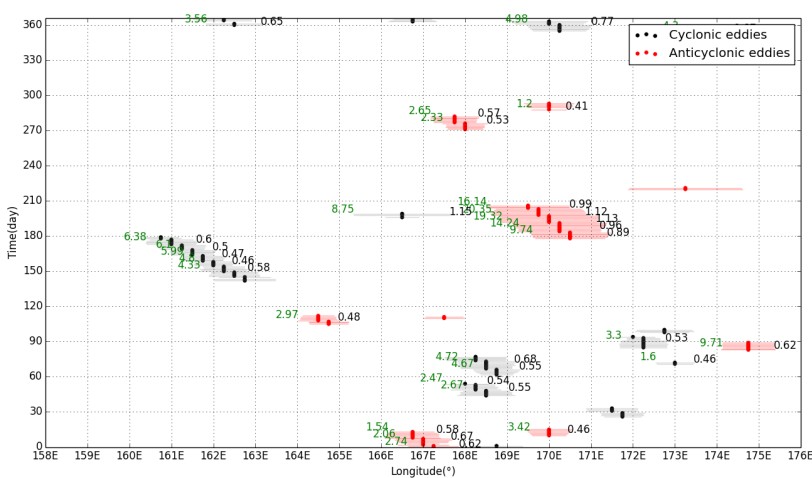

b)





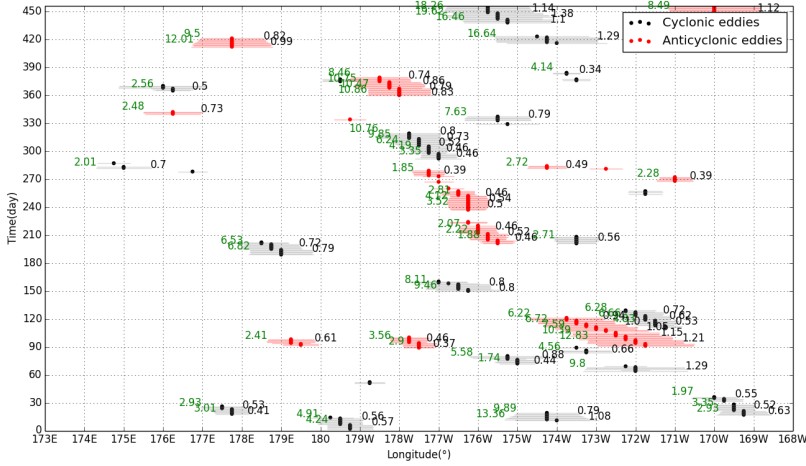


c)

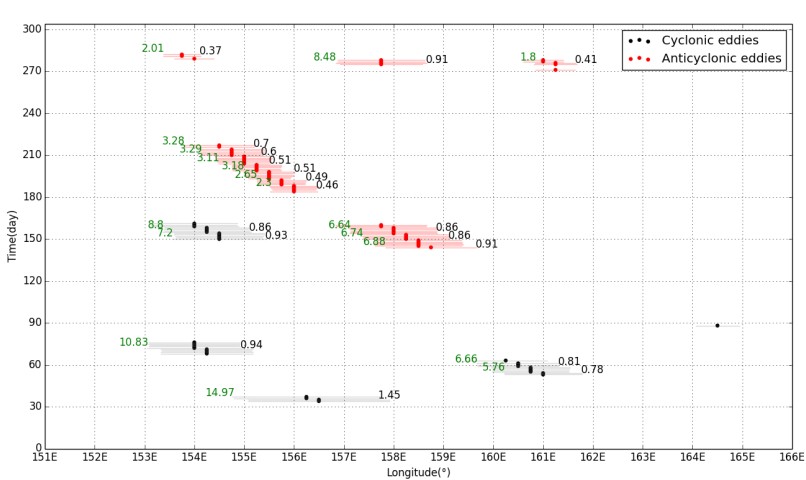

d)





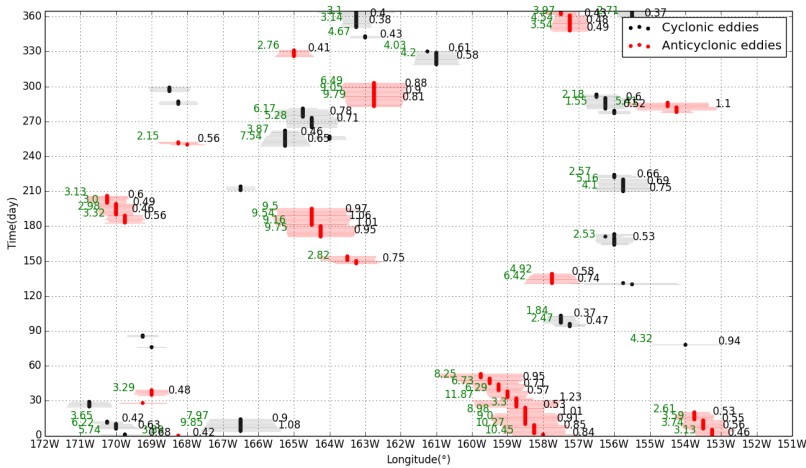

e)

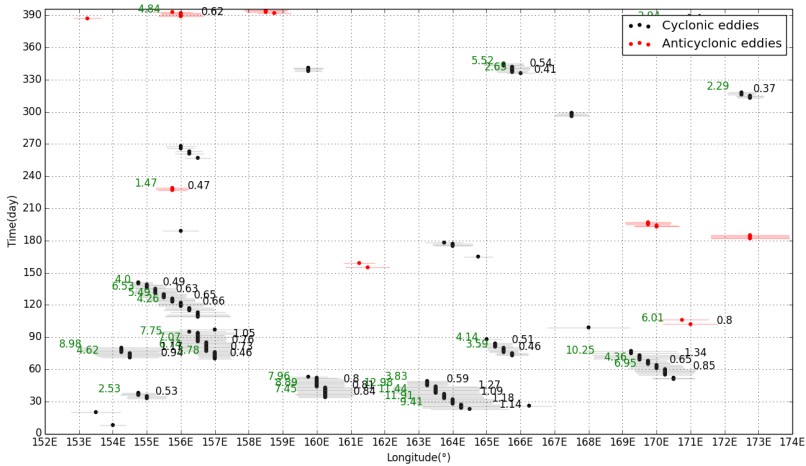

f)





Time-longitude plots for the 6 areas given by the 6 rectangles mentioned in figure 14. The black and the green values (6 days'

interval) indicate the effective radius and the amplitude of the eddy, respectively.

Table 1. Averaged zonal phase speed values of eddy over different time intervals along the given sections at 25 °N in the Central

Pacific Ocean[a]

| Time | Section | Cyclonic | Anticyclonic | Average |
|---|---|---|---|---|
| 2004.05-2005.01 | 158°E-176°E | 5.84 | 3.89 | 4.87 |
| 2004.04-2005.06 | 173°E-168°W | 5.11 | 7.79 | 6.45 |
| 2005.05-2006.03 | 151°E-166°E | 4.17 | 5.01 | 4.59 |
| 2006.02-2007.02 | 172°W-151°W | 2.92 | 4.09 | 3.51 |
| 2007.03-2008.04 | 152°E-174°E | 5.11 | NA | 4.99[b] |
| 2008.03-2008.12 | 165°W-156°W | 4.17 | 3.65 | 3.89 |

[a]Speed values are in cm·s$^{-1}$. NA, not available.

[b]This average value is calculated by cyclonic values only.

Table 2. The statistics of variation of eddy effective radius and amplitude over different time intervals along the given sections at

25 °N in the Central Pacific Ocean[a]

| Time | Section | Cyclonic Inc.: Cyclonic Dec. | AC. Inc.: AC. Dec. |
|---|---|---|---|
| 2004.05-2005.01 | 158°E-176°E | E-Rad. 3:3 Amp. 4:2 | E-Rad. 4:3 Amp. 5:2 |
| 2004.04-2005.06 | 173°E-168°W | E-Rad. 2:3 Amp. 2:4 | E-Rad. 2:6 Amp. 3:5 |
| 2005.05-2006.03 | 151°E-166°E | E-Rad. 1:2 Amp. 2:1 | E-Rad. 1:2 Amp. 2:1 |
| 2006.02-2007.02 | 172°W-151°W | E-Rad. 6:6 Amp. 5:7 | E-Rad. 1:5 Amp. 5:1 |
| 2007.03-2008.04 | 152°E-174°E | E-Rad. 5:4 Amp. 5:4 | NA |
| 2008.03-2008.12 | 165°W-156°W | E-Rad. 4:4 Amp. 3:5 | E-Rad. 1:3 Amp. 3:1 |

[a]Inc., Dec., AC., E-Rad., Amp., are short for increscent, decreasing, anticyclonic, effective radius, amplitude, respectively. NA, not

available.





Figure 16

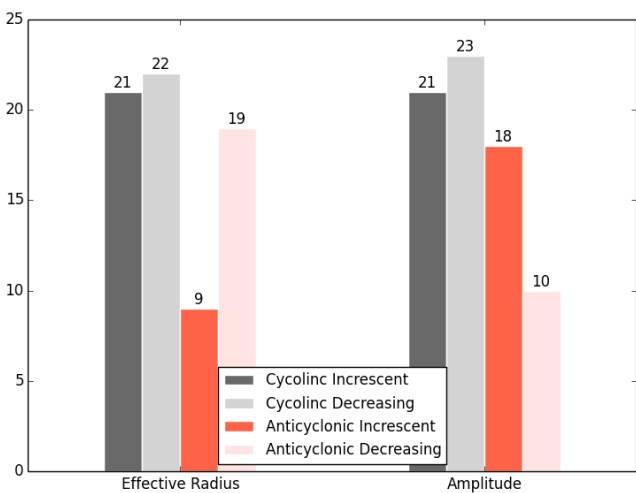


Variation of the effective radius and amplitude of the eddies in the selected zone. Anticyclonic eddies act more unstable than the

cyclonic eddies.

**5 Conclusions**

We have extracted propagation signals of the Rossby wave from MSLA-H fields and made authentication by using SST and

Argo salinity and temperature datasets from 2004 to 2009. We also disposed the results of eddy identification to find out the

propagation features of eddies from 2004 to 2009, and made a comparison between features of the Rossby wave and the eddies.

It's apparent that all the Rossby wave move from the east to the west, but the eddies show different directions of propagation at

the areas influenced by the strong ocean currents. We can conclude that most of evidence indicates that the ocean currents lead the

eddies to migrate (in the zonal direction), and the Rossby wave may plays an accelerative or moderative role in the eddy

propagation. The mechanism of the interaction between the waves and the eddies is still not clear, but the currents are non-

ignorable when analyzing the two phenomena, and further work, including ocean model studies and a detailed comparison of

other attributes which can be influenced by both the two phenomena from other data source, is needed to get better acquainted

with the two fascinating phenomena and the underlying mechanisms.

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

**Acknowledgements**

The Aviso MSLA (-H and -UV), NOAA OI SST and CARDC Argo S/T data can be obtained from http://www.aviso.altimetry.fr/en/data/products/sea-surface-height-products/global.html, http://www.esrl.noaa.gov/psd/data/gridded/data.noaa.oisst.v2.highres.html#detail and http://www.argo.org.cn/index.php?m=content&c=index&f=lists&catid=32, respectively. WOA 2009 data can be obtained from http://www.nodc.noaa.gov/OC5/WOA09/woa09data.html, and OFES zonal velocity data can be downloaded from http://apdrc.soest.hawaii.edu/data/.