# Peer review of "Characteristics of Global Oceanic Rossby Wave and Mesoscale Eddies Propagation from Multiple Datasets Analysis"

_Ocean Science, 2016_

## Editor Comment (EC1) · M. Hecht (Editor) · 29 Sep 2016

Dear Dr. Zhang and co-authors,

As you know, I've had some concern that language issues would present a difficulty in review. This does not necessarily reflect on the scientific content of your paper, but only on the difficulty that potential reviewers may encounter with the English presentation.

In order to proceed with the review process we are requiring a technical language edit. If you choose to arrange for such an editing to be performed, please let me know when you anticipate being able to resubmit a revised version of the manuscript for further consideration. I've informed the journal staff that we will need to extend the Discussion
(review) period to accommodate this additional step.

Sincerely Yours, Matthew Hecht

---

## Referee Comment (RC1) · Anonymous Referee #1 · 8 Dec 2016

Disentangling the Rossby waves from mesoscale eddies, if possible, and understanding their characteristics and influence on each other is an important, unresolved topic. This manuscript uses altimetry tracked SSH anomalies as well as Argo profiles to separately track Rossby waves and eddies. There propagation characteristics are then compared.

One of my primary concerns with the approach taken is here, is how do we know that the authors have successfully separated long wave Rossby waves from the eddies? The authors use an existing product to identify the eddies, which is done via SSH anomaly, but then they also use low pass filtered SSH anomalies to identify Rossby waves. How do we know that these are identifying fundamentally different phenom-

ena? This issue was discussed at length in the Chelton, et al (2011) Progress in Oceanography paper and I don't see anything in this analysis that suggests anything particularly new.

The Argo profile feature tracking is an interesting approach, but it does still raise the same question: why are the observed features identified as long wave Rossby waves and not eddies? In fact, doesn't the propagation of salinity anomaly hint that maybe these are, in fact, eddies that you're identifying as Rossby waves?

These fundamental questions need to be answered before the reader can credibly believe any conclusions about the interaction of the two phenomena. There are some interesting analysis ideas in this manuscript, but I do not see any sufficiently novel results to warrant publication.

---

## Referee Comment (RC2) · Anonymous Referee #2 · 14 Dec 2016

This paper investigates the properties of Rossby waves and mesoscale eddies, based on an analysis of sea surface height supplemented by Argo data, and the interaction between features. The purpose of this paper is interesting, and is certainly appropriate for the journal. The quality of the writing, on the other hand, prevents the results from being adequately communicated for a reader's consideration.

I have read the one review that has already been posted, casting doubt on whether the Rossby waves and mesoscale eddies have been adequately differentiated from one another. This is a serious issue. Beyond this fundamental concern, I simply find it too difficult to understand the precise meaning of much of what is written. A thorough and complete re-write is required. This is beyond the scope of what I would consider to be

a revision. I regret to say that I must recommend against publication.

---

## Editor Comment (EC2) · M. Hecht (Editor) · 15 Dec 2016

Dear Dr. Zhang and co-authors,

as you have seen, Referee #1 has raised a fundamental objection to the lack of clarity in the analytical separation between Rossby waves and eddies, while Referee #2 also objects to the state of the writing. Based on these reviews, I'm very sorry to say that we are rejecting your paper for publication.

You may either formally withdraw the paper at this point, or alternatively we will terminate the review process after the Discussion period ends in January (note that I will be offline and unable to respond from December 17 until January 2).

[Figure]

Again, I'm sorry to deliver this decision. I wish you well with your work.

Sincerely yours, –Matthew Hecht

PS: an approach you might wish to consider in the future, in order to address the difficult problem of delivering a technical paper in fluent English, would be to invite one native English speaker as coauthor. In this case, the objection raised to the lack of unambiguous separation between RW's and eddies might still have been problematic.
* * *